# Instrumented Flexible Glass Structure: A Bragg Grating Inscribed with Femtosecond Laser Used as a Bending Sensor

**DOI:** 10.3390/s23198018

**Published:** 2023-09-22

**Authors:** Loïc Amez-Droz, Matéo Tunon de Lara, Christophe Collette, Christophe Caucheteur, Pierre Lambert

**Affiliations:** 1Department of Aerospace and Mechanical Engineering, Université de Liège, Allée de la Découverte 9, 4000 Liege, Belgium; christophe.collette@uliege.be; 2TIPs Department, CP 165/67, Université libre de Bruxelles, 50 av FD Roosevelt, 1050 Brussels, Belgium; mateo.tunondelara@umons.ac.be (M.T.d.L.); pierre.lambert@ulb.be (P.L.); 3Electromagnetism and Telecommunication Department, University of Mons (UMONS), Boulevard Dolez 31, 7000 Mons, Belgium; christophe.caucheteur@umons.ac.be; 4BEAMS Department, CP 165/56, Université libre de Bruxelles, 50 av FD Roosevelt, 1050 Brussels, Belgium

**Keywords:** compliant mechanism, fused silica, femtosecond laser, Bragg grating sensor

## Abstract

Fused silica glass is a material with outstanding mechanical, thermal and optical properties. Being a brittle material, it is challenging to shape. In the last decade, the manufacturing of monolithic flexible mechanisms in fused silica has evolved with the femtosecond-laser-assisted etching process. However, instrumenting those structures is demanding. To address this obstacle, this article proposes to inscribe a Bragg Grating sensor inside a flexure and interface it with an optical fibre to record the strain using a spectrum analyser. The strain sensitivity of this Bragg Grating sensor is characterized at 1.2 pm/μϵ (1 μϵ = 1 microstrain). Among other applications, deformation sensing can be used to record a force. Its use as a micro-force sensor is estimated. The sensor resolution is limited by our recording equipment to 30 μN over a measurement range above 10 mN. This technology can offer opportunities for surgery applications or others where precision and stability in harsh environments are required.

## 1. Introduction

Manufacturing three-dimensional parts in fused silica with a micrometric resolution is now a reality. The process is performed in two parts. First, a femtosecond laser exposure is applied on the glass substrate following the shape of the designed part. Then, the substrate is placed in a chemical bath for wet etching. Hnatovsky et al. [1] demonstrated that nano-gratings can be introduced in fused silica using a femtosecond laser with various pulse durations, ranging from 40 fs to 500 fs, and the pulse energy threshold to obtain nano-gratings increases with a decreasing pulse duration. Rajesh and Bellouard [2] found the optimal energy deposition value to maximize HF’s fused silica etching rate. They show that the stress generation resulting from the densification of the material is the driving factor in enhancing the etching rate.

Fused silica is a brittle material, its mechanical strength depending mainly on its surface quality, and thus on its manufacturing process. As the intrinsic strength of the Si-O bond is in the order of 21 GPa, approaching this limit can be an opportunity to design highly elastic flexure mechanisms. Bellouard [3] characterized the bending strength of fused silica flexures fabricated by a femtosecond laser (2.7 Gpa). He also demonstrated that a simplified stress model can be used by confirming his estimations using photoelasticity stress measurement inside the flexures. Bellouard et al. [4] also demonstrated that glass manufacturing by femtosecond-laser-assisted etching is suitable for high-demanding technologies such as optofluidics, optomechanics, marking and photonics. Later, Athanasiou and Bellouard [5] proposed a monolithic structure to perform tensile tests on a fused silica sample without the necessity to mechanically interact with the specimen. They used what they call stressors to load the specimen using the femtosecond laser. Laser-affected zones exhibit a net volume expansion and thus stress. They reached stress levels up to 2.4 Gpa. They have also studied the brittle-fracture statistics of fused silica under femtosecond laser exposure [6]. They have found a method to extract the Weibull statistics by following the apparition of chaotic patterns along exposed lines where periodic nano-gratings are expected. This defect can result from different phenomena such as plasma dynamic relaxation, the excitons dynamics, the crack nucleation rate of pulses as well as the scanning velocity. Secondary is also to consider including the shockwave propagation and the residual loads. Recently, Casamenti et al. [7] optimized the femtosecond-laser-assisted etching process by tuning the energy dose and the use of various etchants. They even demonstrated that using NaOH, a less-dangerous etchant, enables a higher etching rate than currently used HF and KOH resulting in higher aspect ratio manufacturing. Concerning the possible enhancement of fused silica part mechanical strength, Widmer et al. [8] demonstrated that their strength can be increased by annealing. By annealing the part at 1200 ${}^{\circ }$C, it resulted in a mechanical strength increase from 8 GPa to 10 GPa due to the surface stress concentration reduction. These tests were performed by applying compressive stress on the specimens.

Fused silica has strong temperature stability in terms of dilatation (∼0.5 ppm/${}^{\circ }$C). However, for precision mechanisms, its temperature coefficient of elasticity is not negligible (∼220 ppm/${}^{\circ }$C). Vlugter and Bellouard [9] demonstrated that irradiating amorphous silica with a femtosecond laser can decrease its temperature coefficient of elasticity beyond 50%.

Femtosecond-laser-assisted etching is not the only method to manufacture micro-scale glass structures. Toombs et al. [10] presented an additive manufacturing process of fused silica. A photopolymer-silica nanocomposite is illuminated tomographically. Then, the substrate is sintered to result in a fused silica part. The minimum feature size is 20 μm with a surface roughness of 6 nm.

Fused silica flexure mechanisms have been design for various applications. Bellouard et al. [11] presented a monolithic optomechanical micro-displacement sensor (50 nm resolution). The sensitive element is translational guidance containing an integrated linear encoder. The displacement is measured optically through an integrated waveguide in the sensor frame that is opened facing the linear encoder perpendicularly. Lenssen and Bellouard [12] demonstrated a transparent glass monolithic micro-actuator. The actuation is performed by a capacitive comb-array. A transparent conductive material (indium-tin-oxide layer) is deposed on the structure. The comb array is guided in translation by a flexure mechanism. Nazir and Bellouard [13] proposed a laser-to-fibre coupling mechanism contained in a monolithic fused silica tunable flexure mechanism (sub-nm resolution positioning over a tens of micrometres range of motion). Nazir et al. [14] also used their concept of stressors to load a micro-tensile tester to perform long static stress measurements under normal atmospheric conditions. Zanaty et al. [15] presented a multistable glass monolithic mechanism to perform safe surgical puncturing. The force and stroke applied by the needle at the output of the mechanism are decoupled from the operator input and are tunable (5–20° stroke with a force exceeding 8 mN). Tissot-Daguette et al. [16] presented a constant-force surgical tool based on a monolithic glass flexure mechanism (1 gf with a 0.1 gf maximum variation over a displacement range of 870 μm). A microscope is used as a visual feedback of the output displacement.

The femtosecond laser can also be used to change the optical properties of fused silica. Drevinskas et al. [17] used a femtosecond laser to fabricate birefringent surface elements in fused silica. They demonstrated that the written nano-gratings exhibit up to a three-fold increase after polishing and 25 h of KOH etching at room temperature. This led to achromatic behaviour over the entire visible spectral range, enabling opportunities for micro-optics applications. Beresna and Kazansky [18] manipulated induced form birefringence in fused silica using a femtosecond laser to obtain circular-polarization beam splitters. The birefringence results from the spontaneous formation of subwavelength periodic structures oriented perpendicular to the writing beam polarization in the focal volume. Richter et al. [19] studied the formation of the nano-gratings in fused silica using a femtosecond laser. They have demonstrated that they can resist extremely large temperatures up to 1150 ${}^{\circ }$C. Also, they were able to tune precisely their form birefringence to manufacture phase elements such as quarter- and half-wave plates. Most recently, Radhakrishnan et al. [20] observed that localized permanent densification can be obtained between non-overlapping simultaneous femtosecond-laser-affected zones while preserving the integrity of the material. This method can be used for non-contact laser-induced high-pressure studies (a few tens of GPa).

Zhang et al. [21] demonstrated a method to integrate strain-optic and thermo-optic Bragg grating sensors in bulk fused silica using an ultrafast laser. Their Bragg grating sensors are thermally stable to 500 ${}^{\circ }$C.

From this study, our goal is to instrument a flexible glass structure by integrating a Bragg grating sensor. Nowadays, the deformation of a structure is monitored by joining an optical fibre Bragg grating (FBG) sensor to it. This is well known, but can be quite bulky depending on the application. A conventional single-mode optical fibre has a cladding diameter of 125 μm. By integrating the sensor directly inside the structure, the overall dimension can be reduced. Therefore, the stiffness of the structure can be reduced. We reported a method to write a waveguide in a glass substrate [22]. It is the first step in our lab to enable readout opportunities for Bragg grating sensor integration in various three-dimensional structures. Then, our studies on Bragg grating sensor manufacturing inside a micro-scale fused silica structure are reported in [23]. The sensitivity to temperature and to axial strain in a tensile specimen of the Bragg grating sensor is obtained.

This paper presents the integration of a Bragg grating sensor inside fused silica structures for bending monitoring. First, two different bending experiment configurations are performed: in a three-point flexural test and by loading the tip of a cantilever beam. The strain sensitivity of the Bragg grating sensor can be compared to the FBG theory and to [21]. Then, to validate our approach for instrumenting micro-scale mechanisms, the Bragg grating sensor is inscribed in a 50 μm thick flexure. The cross-spring pivot hinge design of Tielen and Bellouard [24] has been selected. This joint is well-suited for large rotational guiding and a Bragg grating sensor can be placed inside to measure its deflection.

The fabrication of the specimens and their Bragg gratings are explained in Section 2. The strain sensitivity is described depending on the three specimen geometries (three-point flexural test, cantilever beam and flexure pivot joint lever) and their loading method. Then, in Section 3, the mechanical estimation of the applied strain is shown in function of the Bragg grating wavelength shift for each experiment and compared to the theoretical strain sensitivity of a Bragg grating. Finally, Section 4 presents the possibility of using the demonstrated method to design a force sensor and how it can be used for other glass structure instrumentations.

## 2. Materials and Methods

The Bragg grating sensors as well as the specimen manufacturing are obtained using a femtosecond laser included in a machine called FEMTOprint (Ramat Hasharon, Israel). This machine features a three-axis precision moving platform (100 nm resolution) on which the glass substrate is fixed. The UV-grade fused silica glass substrate is provided by Siegert Wafers GmbH (Aachen, Germany). A Thorlabs LMH-20x-1064 objective (Newton, NJ, USA) is used to focus the laser. The voxel waist is 1.5 μm in diameter and its height is 24 μm. To define the laser toolpath, a modified version of Alphacam by FEMTOprint SA is used. Each parameter of the laser can be tuned independently. After the laser exposition, the substrate is placed in a 12 M KOH bath at 85 ${}^{\circ }$C for wet etching.

The waveguide and the Bragg grating are inscribed at first. The Bragg grating shall be placed as far away as possible from the neutral beam axis to maximize its sensitivity to the beam bending. The neutral beam length, located at the centre thickness (dotted line in Figure 1c), is constant during bending. For further notation, the length of the Bragg grating is noted $L_{s}$ and the length of the neutral axis in the portion of the beam where the Bragg grating is present is noted $L_{0}$. The strain applied to the Bragg grating in a horizontal portion of the beam at a horizontal position *x* can be estimated using the Euler–Bernoulli equation as in [25]:(1)$$\epsilon \left(x\right)=\frac{\Delta L}{L}=\frac{\alpha \rho (x)-\alpha (\rho (x)+y)}{\alpha \rho (x)}=\frac{y}{\rho (x)}=\frac{M(x)y}{EI}$$
using *L*, the length of the portion of the beam (1.5 mm), α, the bending angle, ρ, the bending radius, *y*, the position of the centre of the Bragg grating sensor from the neutral axis (10 μm), *M*, the bending moment, *E*, the Young modulus of fused silica (72 GPa) and *I*, the inertia of the considered beam section. Considering the beam is thin enough (its thickness *h* = 50 μm is much smaller than its length *L*), shear stress can be neglected with respect to normal stress. Therefore, the Euler–Bernoulli equation relates the loading of the beam expressed in terms of bending moment *M* to the geometrical deformation expressed by a local curvature radius ρ ($\frac{1}{\rho }=\frac{M}{EI}$).

The Bragg grating is fabricated in the substrate by inducing a periodic defect plane-by-plane with the femtosecond laser [23]. The planes are perpendicular to the inscribed waveguide. They are made of lines with the femtosecond laser. Its translation speed is 15 mm/min and the pulse energy is 150 nJ with a repetition rate of 1 MHz. The Bragg grating wavelength is the centre of gravity of the reflected spectrum defined as
(2)$$\lambda_{\mathrm{BG}}=2n_{\mathrm{eff}}\frac{\Lambda }{m}=1595\;\mathrm{nm}$$
with $n_{\mathrm{eff}}$ being the effective refractive index of the mode transmitted in the waveguide (∼1.45), Λ being the period of the defect (∼1.1 μm) and *m* being the order of the mode (2). The waveguide is obtained by writing parallel lines along the direction of the desired optical path with the femtosecond laser. The step between the laser paths is 200 nm. The translation speed of the laser is 20 mm/min with a pulse energy of 130 nJ with a repetition rate of 1 MHz. Our previous work [22] reports the intensity loss of the waveguide of 2.2 dB/cm.

The Bragg grating is sensitive to axial strain and to the temperature gradient. The shift of the central wavelength of the Bragg grating reflected amplitude spectrum can be expressed as
(3)$$\Delta \lambda_{\mathrm{BG}}=2\left(\Lambda \frac{dn_{\mathrm{eff}}}{dT}+n_{\mathrm{eff}}\frac{d\Lambda }{dT}\right)\Delta T+2\left(\Lambda \frac{dn_{\mathrm{eff}}}{d\epsilon }+n_{\mathrm{eff}}\frac{d\Lambda }{d\epsilon }\right)\Delta \epsilon$$
with ΔT, the temperature gradient and Δϵ, the axial strain.

For this study, the temperature gradient is considered negligible compared to the applied strain during the experiments. Our previous work [23] reported a temperature-optic sensitivity of 10.5 pm/${}^{\circ }$C. The duration of the test is less than 20 min. A temperature gradient of less than 1 ${}^{\circ }$C is assumed. The temperature influence on the Bragg grating is in the same order of magnitude as the data acquisition system FiberSensing BraggMETER FS 2100 (5 pm resolution). As described in [26], the sensitivity to axial strain can be expressed as
(4)$$\frac{\Delta \lambda_{\mathrm{BG}}}{\Delta \epsilon }=\lambda_{\mathrm{BG}}\left(1-p_{e}\right)$$
with $p_{e}$, the strain-optic constant of the material. For fused silica, $p_{e}$ can be defined as in [27]:(5)$$p_{e}=\frac{n_{\mathrm{eff}}^{2}}{2}\left[p_{12}-\nu \left(p_{11}+p_{12}\right)\right]$$
with $p_{11}$ and $p_{12}$, the elasto-optic independent coefficients for bulk silica (0.121 and 0.270), listed in [28], ν and $n_{\mathrm{eff}}$, respectively, its Poisson’s ratio and the effective refractive index (0.16 and 1.45).

With the defined $\lambda_{\mathrm{BG}}\approx 1595$ nm, it gives a theoretical sensitivity to an axial strain of 1.24 pm/μϵ. μϵ corresponds to a strain of $10^{-6}$. It is suited to express typical strain values for Bragg grating sensors.

This sensitivity is then compared to the characterized sensitivity deduced from the following experiments by estimating the applied strain from mechanical assumptions. The Bragg grating shift is measured by interfacing the Bragg grating of the specimen to the data acquisition system FiberSensing BraggMETER FS 2100 with an optical fibre.

To verify the operating principle of our Bragg grating bending sensor, three experiments are performed. These tests allow us to characterize the strain sensitivity of the sensor. The first one, the three-point flexural test, is a conventional test to obtain the stress–strain response of a material. The second test, the cantilever beam bending test, verifies the conventional beam deflection model in structural engineering. Finally, the third test, the flexure pivot joint lever, verifies the sensor integration in a micro-scale flexure to validate its use in monolithic micro-scale mechanism design.

### 2.1. Three-Point Flexural Test

In this first experiment detailed in Figure 1, the load is applied at the centre of the beam. The beam is horizontal and sits on two fixed rods. The bending moment can be expressed as
(6)$$M\left(x\right)=\frac{F}{2}x\;x\in \left[0;\frac{L}{2}\right]$$
with *F*, the loading at the centre of the beam, *x*, the position from one fixed point and *L*, the distance between the two fixed points.

The force is applied with a stainless steel cantilever beam clamped on a vertical translation moving stage. A 3.2 mm diameter ceramic ball is glued at the tip of this cantilever beam to apply the load at a precise position on the glass beam. The displacement of the vertical stage $f_{1}$ and the depth at the ball location $f_{2}$ are measured using two laser displacement sensors Keyence LC-2440 (Champaign, IL, USA) (1 μm resolution). The stiffness $K_{\mathrm{fF}}$ of the stainless steel cantilever beam is obtained experimentally. This calibration is performed by placing a fixed precision scale OHAUS YA102 (Dundas, ON, Canada) (10 mg resolution) under the ceramic ball and applying the load by translating the vertical stage.
(7)$$K_{\mathrm{fF}}=\frac{F_{\mathrm{scale}}}{f_{1}}$$Then, the load applied on the glass beam can be expressed as
(8)$$F=K_{\mathrm{fF}}\left(f_{1}-f_{2}\right)$$The Bragg grating is 5 mm long and is located at the horizontal centre of the beam ($x=\frac{L}{2}$) at a distance *y* from the neutral axis (83 μm). Its length being much smaller than *L* (30 mm), we can assume
(9)$$\epsilon =\frac{K_{\mathrm{fF}}Ly}{4EI}\left(f_{1}-f_{2}\right)$$
with *I*, the inertia of the considered beam section, $I=\frac{bh^{3}}{12}$ with *b*, the beam width (4 mm) and *h*, the beam thickness (0.5 mm).

To allow for characterizing the sensitivity of the Bragg grating to strain in traction and compression, the beam can be flipped so the Bragg grating is in the opposite direction from the neutral axis.

### 2.2. Cantilever Beam

In this experiment detailed in Figure 2, the load is applied at the tip of the cantilever beam. The beam is horizontal and clamped on one side. The bending moment can be expressed as
(10)M(x)=F(L−x)x∈[0;L]
with *F*, the loading at the tip of the beam, *x*, the position from the clamp and *L*, the distance between the clamp and the position where the load is applied.

The loading is applied in the same way as the three-point flexural test. To allow characterizing the sensitivity of the Bragg grating to strain in traction and in compression, the beam can be returned so the Bragg grating is in the opposite direction from the neutral axis. The Bragg grating is 5 mm long and is located close to the clamp position (x=0) at a distance *y* from the neutral axis (83 μm). Its length being much small than *L* (∼24 mm), we can assume
(11)$$\epsilon =\frac{K_{\mathrm{fF}}Ly}{EI}\left(f_{1}-f_{2}\right)$$
with *I*, the inertia of the considered beam section, $I=\frac{bh^{3}}{12}$ with *b*, the beam width (4 mm) and *h*, the beam thickness (0.5 mm).

### 2.3. Flexure Pivot Joint Lever

In this experiment detailed in Figure 3, the load is applied at a distance *r* from the centre of the pivot on a needle on the free end of the flexure joint. This pivot is a cross-spring hinge. It is composed of 3 beams of length *L*. The central beam has a width of *b*. The other two beams are perpendicular to the central one and have a width of b/2. The Bragg grating is written along the whole length of the centre beam at a distance *y* from its neutral axis. With the lever arm *r* being much larger than the beam length, the bending moment along the beam can be considered constant. Therefore, the radius of curvature can be expressed as ρ=L/α, α being the bending angle, and the strain as
(12)$$\epsilon =\frac{\Delta L}{L}=\frac{\alpha \rho -\alpha (\rho +y)}{\alpha \rho }=\frac{y}{\rho }=\frac{y}{L}\alpha$$

The pin used to load the specimen is a leg of a through-hole electronic component attached to a translation stage perpendicular to the needle at its rest position. To allow characterizing the sensitivity of the Bragg grating to strain in traction and in compression, the pin can push or pull the needle. The bending angle can be expressed as
(13)$$\alpha \approx \tan\alpha \approx \frac{d}{r}$$
with *d*, the displacement of the translation stage.

*r* value is calibrated using the graduation on the fixed part of the specimen close to the tip of the needle. *d* is measured using a laser displacement sensor Keyence LK-H022 (Champaign, IL, USA) placed in front of the translation stage. The data are recorded using an NI USB-6002 (Austin, TX, USA) analogic input connected to a laptop with a dedicated LabVIEW platform. The Bragg grating reflected spectrum is recorded synchronously with the same LabVIEW platform using a serial ethernet connection.

## 3. Results

The sensitivity characterization experiments of the Bragg grating of our different setups are reported in Figure 4. Depending on the experiment, the strain testing range is different. It has been chosen for each to limit the probability of breakage. The characterized sensitivities of the Bragg grating of the three designs are in good agreement with the theoretical sensitivity (1.24 pm/μϵ). The sensor manufacturing is therefore validated and further tests can be organized in different configurations and in various environments depending on the application.

## 4. Discussion

This study has demonstrated a method to design instrumented glass structures. In terms of application, it can improve the solutions where a fibre Bragg grating sensor needs to be integrated into a structure. Our first study [23] presenting the temperature-optic and strain-optic sensitivities verified the Bragg grating writing process in planar fused silica substrates. The strain-optic characteristics were obtained through a tensile experiment similar to the FBG sensor testing. For the integration in monolithic flexure mechanisms, the sensitivity of this sensor needs to be characterized in a flexible structure. The work of Zhang et al. [21] presented the first integration of a Bragg grating sensor in a 1 mm thick fused silica substrate. A three-point flexural test was performed to assess the sensitivity of the sensor. They have obtained strain-optic sensitivities from 1.15 pm/μϵ to 1.38 pm/μϵ. Our results close to the theoretical strain-optic sensitivity obtained with the same type of experiment confirm the validity of our process. Moreover, the possibility presented in this work to write the sensor inside a micro-scale structure allows us to reduce the size of the sensitive structure, and thus, its stiffness. It is also a gain in terms of production and assembly complexity as the sensor is fully monolithic. Besides this application, it is now also possible to instrument existing glass flexure mechanisms such as the constant-force surgical tool presented by Tissot-Daguette et al. [16]. It would enable estimating the displacement of the tool’s tip through the instrument.

As an example of a force sensor, our structure presented in Figure 3 can be used. By the bending angle Bragg grating strain relationship from Equation (Equation 12) ($\alpha =\frac{L}{y}\epsilon$), a force measured perpendicularly to the needle at its tip (at *r* = 10 mm) can be estimated as
(14)$$F=\frac{M}{r}=\frac{K_{\alpha M}\alpha }{r}=\frac{Ebh^{3}}{6ry}\epsilon$$
with *M*, the bending moment of the pivot joint and $K_{\alpha M}=\frac{Ebh^{3}}{6L}$, its bending stiffness with *b*, the width of the central beam (200 μm), the other two beams have both a width of b/2 and *h*, the thickness of the beam (50 μm).

From the theoretical strain sensitivity of the Bragg grating $\frac{\epsilon }{\Delta \lambda_{\mathrm{BG}}}$ = 1.24 pm/μϵ, the force sensitivity can be expressed as
(15)$$\frac{F}{\Delta \lambda_{\mathrm{BG}}}=\frac{Ebh^{3}}{6ry}\frac{\epsilon }{\Delta \lambda_{\mathrm{BG}}}=2.4\;\mu N/\mathrm{pm}$$
Considering that the resolution of our optical spectrum analyser is $\Delta \lambda_{\mathrm{BG},\min}$ = 5 pm, the force resolution of this sensor could reach $\Delta F_{min}$ = 12 μN. By taking into account the standard deviation of the experiment, the force resolution is limited to $\Delta F_{min}$ = 30 μN. It is close to the theoretical limit of the spectrum analyser. To improve this resolution, the Bragg grating can be placed closer to the surface of the beam. Being 10 μm width, and by placing it at 5 μm from the surface instead of 10 μm, its position from the centre *y* could be increased by a factor of 1.5. Also, it is also possible to select an optical spectrum analyser with a better wavelength resolution (existing down to 1 fm). By taking into account the fused silica strength (>1 GPa), the force measurement range is >10 mN. This sensor has three orders of magnitudes between its measurement range and its resolution.

Further studies will focus on the use and characterization of this type of sensor in harsh environments such as in water and with temperature gradients. The decoupling of the temperature and strain sensitivities is then the point of interest of these studies. Considering instrumenting multiple degrees of freedom sensors is also a potential for complex system analysis. As an example, the two degrees of freedom force sensor presented by Buttafuoco et al. [29] can be obtained in glass and instrumented with Bragg grating sensors inscribed in the flexures.

## 5. Conclusions

Instrumenting a glass flexure with an inscribed Bragg grating sensor has been demonstrated. The same femtosecond laser has been used to manufacture the monolithic mechanism and the sensor. The correlation of our Bragg grating sensors’ strain sensitivity with the theoretical sensitivity validates our designs and the manufacturing method. This technology is then available to instrument glass mechanisms.

The next step is to assess the operating range of such kinds of sensors according to the outstanding properties of fused silica glass.

## Figures and Tables

**Figure 1 sensors-23-08018-f001:**
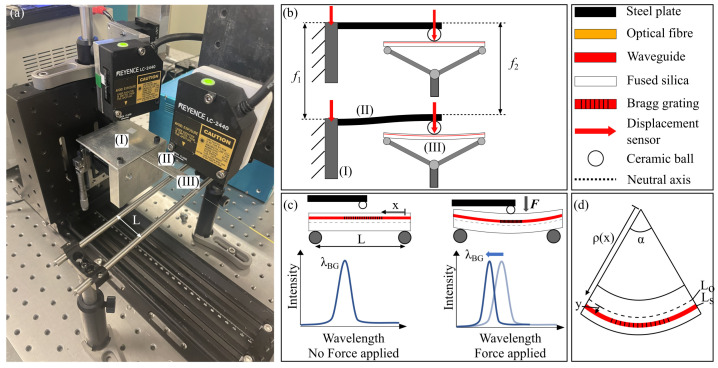
(**a**) Three-point flexural test bench. (**b**) A stainless steel beam with a ceramic ball is used to load the glass test beam. Two displacement sensors are used to monitor the stainless steel clamped side displacement $f_{1}$ and its ball end displacement $f_{2}$. (**c**) The test beam is placed on two fixed rods. The load is applied from the top at the centre of the beam. The Bragg grating sensor is placed close to the top surface of the beam, the applied strain is in compression. Therefore, the Bragg grating wavelength shift is negative. (**d**) It is a close view of the centre of the beam showing where the Bragg grating is inscribed.

**Figure 2 sensors-23-08018-f002:**
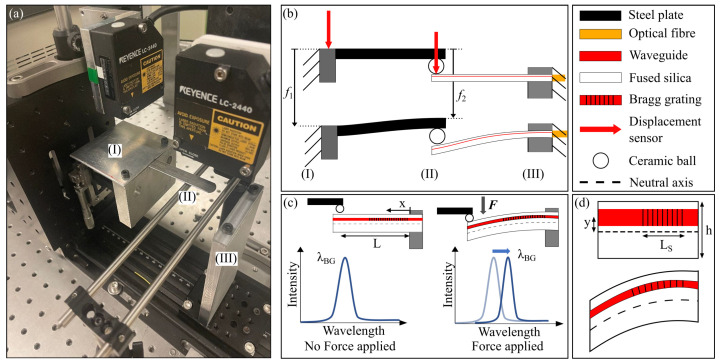
(**a**) Glass cantilever beam test bench. (**b**) A stainless steel beam with a ceramic ball is used to load the glass test beam. Two displacement sensors are used to monitor the stainless steel clamped side displacement $f_{1}$ and its ball end displacement $f_{2}$. (**c**) The test beam is clamped on the optical fibre side. The load is applied from the top at the tip of the cantilever beam. The Bragg grating sensor is placed close to the top surface of the beam, the applied strain is in compression. Therefore, the Bragg grating wavelength shift is negative. (**d**) It is a close view of the beam showing where the Bragg grating is inscribed close to the beam clamp position.

**Figure 3 sensors-23-08018-f003:**
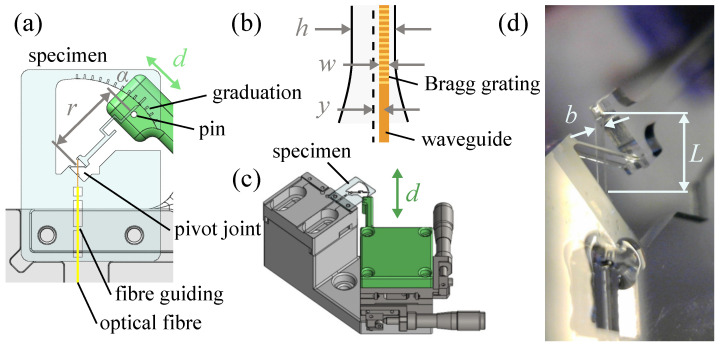
(**a**) Flexure specimen containing a cross-spring pivot hinge which has a needle being part of its moving end. A pin is applied perpendicular to the needle at a distance *r* of the centre of the pivot. The graduation monitors the bending angle α. (**b**) The Bragg grating is included in a waveguide of width *w* and is placed at a distance *y* from the neutral axis of the central beam. The beam has a thickness of *h*. (**c**) The specimen is placed on a test bench such as the pin facing the needle. The needle is pushed by the pin by moving the translation stage manually. The displacement of the pin is expressed as *d*. (**d**) The close view represents the glass pivot joint lever. Its central beam has a width *b* = 200 μm and a length *L* = 1.5 mm.

**Figure 4 sensors-23-08018-f004:**
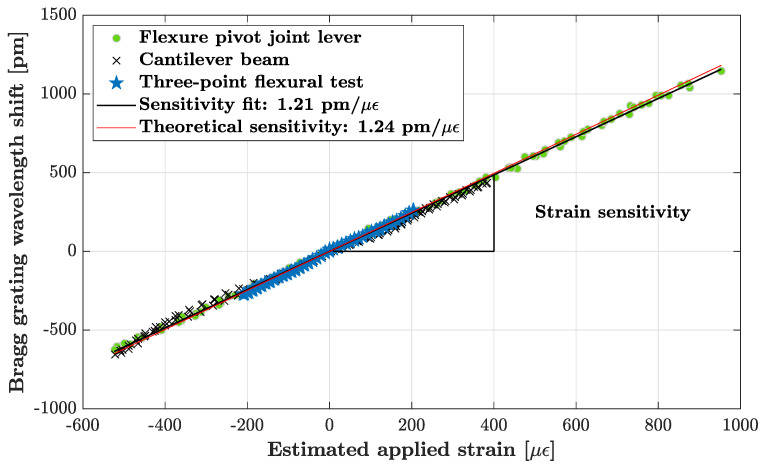
Results of the characterized Bragg grating sensitivity to strain for the three specimen designs that are detailed in Figure 1, Figure 2 and Figure 3. Each experiment has been performed 4 times. The standard deviation of the experiments from the linear fit is 10 μϵ.

## Data Availability

The data that support the findings of this study are available from the corresponding author upon reasonable request.

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
