# Peer review of "Instrumented Flexible Glass Structure: A Bragg Grating Inscribed with Femtosecond Laser Used as a Bending Sensor"

_sensors, 2023, doi:10.3390/s23198018_

Round 1
Reviewer 1 Report
The authors investigate a force/displacement sensor fabricated in silica glass. The sensor consists in a flexure with a Bragg grating embedded in its inner structure, above the neutral axis of deformation. As the flexure bends, the Bragg grating senses the localized deformation, which, through modelling, can be correlated to both, displacement and force applied to the flexure. The flexure and the Bragg grating are manufactured using the same machining process and in a single monolith.
The paper provides a convincing demonstration on how femtosecond laser technologies can be used to fabricate a complete sensor that includes both mechanical and optical elements. As such, it fits well within the scope of this journal. It forms an innovative contribution to the existing literature.
We recommend publishing it, pending the authors address the following comments:
1/ Very few details are provided about the Bragg grating. What is the estimated index contrast? How long is the grating? What is its reflectance? How lossy is it? Are the individual lines forming the grating homogeneous?
2/ Likewise, more information should be provided about the waveguide written in the flexure. What is the refractive index contrast? We assume that the waveguide is single mode. Is it correct?
3/ What about the natural frequency of the flexure?
4/ The author could also further discuss how to expand the number of degrees of freedom of such structure.
In general, the paper is well written and clear. There are a few typos and inor edits that should be fixed. ('et al.' being Latine should be in italic.)
Reviewer 2 Report
In this manuscript, the author presents a solution for strain sensing using femtosecond laser-etched Bragg gratings. The functionality of this sensor is demonstrated through a series of three tests employing distinct mechanical designs. The outcomes of these experiments reveal a strain sensitivity of 1.21 pm/µÏµ, closely aligned with the theoretically predicted 1.24 pm/µÏµ. The author provides a theoretical model to explain the Bragg grating and a mechanical model to describe the strain generated in the experiment. This draft holds the potential for publication, provided that certain inquiries are adequately addressed:
-
1. The method by which the Bragg grating wavelength is detected remains unspecified within the draft.
-
2. What is the impact of temperature variations on sensitivity? Within which range could the impact be ignored in your design?
-
3. Given that the Bragg grating is situated off the primary axis, what is the quantitative influence of this displacement on strain sensing sensitivity?
Reviewer 3 Report
In this research addressed Bragg Grating sensor inside a flexure and interface it with an optical fiber to record the strain using a spectrum analyzer. The strain sensitivity of this Bragg Grating sensor is characterized at 1.2 pm/µÏµ. The work is well presented, and the idea is clearly presented. Some minor points only for authors before it is published:
1- English typo in some sentences, please revise it.
2- A full discerption for the used instruments should be address.
3- References of the Euler-Bernoulli equation.
4- The novelty of the work should be addressed, furthermore, a comparison between this work and published work should be presented.
5- Please add the spectrum curve to allow the reader to understand the idea.
Need some improvments.
Round 2
Reviewer 3 Report
The authors review all the comments, so, accepted in the current format.